# Contextualizing Parental/Familial Influence on Physical Activity in Adolescents before and during COVID-19 Pandemic: A Prospective Analysis

**DOI:** 10.3390/children7090125

**Published:** 2020-09-03

**Authors:** Barbara Gilic, Ljerka Ostojic, Marin Corluka, Tomislav Volaric, Damir Sekulic

**Affiliations:** 1Faculty of Kinesiology, University of Split, 21000 Split, Croatia; bargil@kifst.hr (B.G.); ljerka.ostojic@mef.sum.ba (L.O.); 2Faculty of Kinesiology, University of Zagreb, 10000 Zagreb, Croatia; 3Faculty of Medicine, University of Mostar, 88000 Mostar, Bosnia and Herzegovina; 4Faculty of Science and Education, University of Mostar, 88000 Mostar, Bosnia and Herzegovina; marin.corluka@fpmoz.sum.ba (M.C.); tomislav.volaric@fpmoz.sum.ba (T.V.)

**Keywords:** parenting, crisis, risk factors, protective factors, puberty, social distancing

## Abstract

Parental and familial factors influence numerous aspects of adolescents’ lives, including their physical activity level (PAL). The purpose of this study was to evaluate the changes in PAL which occurred during the COVID-19 pandemic, and to evaluate influence of sociodemographic and parental/familial factors on PAL levels before and during pandemic in adolescents from Bosnia and Herzegovina. The sample included 688 adolescents (15–18 years of age; 322 females) who were tested on two occasions: in January 2020 (baseline; before the COVID-19 pandemic) and in April 2020 (follow-up; during the COVID-19 pandemic lockdown). Variables included PAL (measured by the Physical Activity Questionnaire for Adolescents–PAQ-A) as well as sociodemographic-, parental-, and familial factors. A significant decline in PALs was recorded between baseline and follow-up (*t*-test: 11.88, *p* < 0.001). Approximately 50% of adolescents underwent sufficient PAL at baseline, while only 24% of them were achieving sufficient PAL at the time of follow-up measurement. Paternal education was positively correlated (OR (95%CI): baseline: 6.63 (4.58–9.96), follow-up: 3.33 (1.19–7.01)), while familial conflict was negatively correlated (baseline: 0.72 (0.57–0.90), follow-up: 0.77 (0.60–0.99)) with PALs before and during the pandemic. This study highlights the importance of the parent–child relationship and parental/familiar support in promoting physical activity both during regular life and during crises and health challenging situations like the COVID-19 pandemic.

## 1. Introduction

Physical activity is defined as any movement of the body that results in energy expenditure. It can be categorized as occupational, transportational, sports activity, household activities, and leisure-time activities [1]. Maintaining an adequate physical activity level (PAL) is essential for sustaining and improving metabolic, and psychological functions, as well as the overall health and life quality of a human being [2]. Physical activity has a positive influence on the immunological system and can decrease the incidence of communicable diseases such as bacterial and viral infections [3]. However, due to technological advancement, physical activity in modern life has drastically reduced, directly contributing to the development of many non-communicable diseases (e.g., diabetes, cardiovascular diseases, obesity) [4].

Adolescence is considered to be a critical period with regard to maintaining appropriate PAL over the lifespan. First, the trend of PAL decline during aging is evident, with the greatest decrease evidenced during adolescence [5,6]. Globally, 77.6% of boys and 84.7% of girls aged 11 to 17 years achieve an inadequate PAL [7,8,9]. Second, considering the fact that low PAL in childhood and adolescence has numerous severe health consequences, movement habits attained during this life period impact the maintenance of adequate PAL later in adulthood [10]. Not surprisingly, a great number of studies have investigated the trends of changes in PAL, and factors that influence such trends, with the intention to develop precise and adequate interventions for maintaining/improving PALs in adolescence [11,12].

At the beginning of the year 2020 there was a rapid expansion of the COVID-19 virus, and by March 11, a pandemic had been declared [13]. Countries all around the globe imposed several measures for stopping and slowing down the spread of the disease. Since the virus spreads by saliva droplets, one of the most frequently employed measures was so-called “social distancing” [14]. In general, social distancing is the practice of maintaining a greater than usual physical distance (2 m) from other people or avoiding direct contact with people or objects in public places. The main intention is to minimize exposure and reduce the transmission of infection. In the real world, it meant that kindergartens, schools, universities, sports clubs, and fitness centers were closed. Due to the mentioned measures and related to the movement restrictions and closing of sport and recreation facilities, a decline in PAL was expected and evident [15]. Concerning the fact that there is generally inadequate PAL under normal life circumstances, this kind of movement limitation was naturally considered to have serious health consequences and possibly impair future healthy life habits [16]. Indeed, studies clearly pointed to a significant decrease in PAL as a result of COVID-19-induced measures of social distancing [17,18].

Factors such as gender, age, social support from parents and peers, parental education, motivation, self-esteem, knowledge of exercising, and the environment impact the PAL among children and adolescents [19,20,21]. It would be logical to expect that similar factors influence the PAL during crises like the COVID-19 pandemic. Supportively, recent studies have shown that the pre-pandemic fitness status, gender, and living environment are factors that influenced changes in PAL during the COVID-19 pandemic [17,18,22]. Specifically, a higher fitness status has been associated with a higher PAL before and during the pandemic in Croatian adolescents [17], and a greater PAL decline has been noted for boys in comparison with girls [17]. Similar conclusions were provided in an Italian study [18]. Furthermore, the PALs of adolescents living in urban environments declined more than those of their rural peers due to the closure of sport facilities in urban communities as well as the lower pre-COVID PALs in rural adolescents [22].

Family factors such as parental education, parental social support, and family structure, greatly impact the health-related habits of children, including their PALs [23,24]. Specific to the COVID-19 pandemic, Moore et al. recently reported that parental encouragement and parental co-participation are highly associated with healthy movement behaviors of Canadian children and adolescents during the COVID-19 pandemic [25]. Indeed, parents have an extremely important role in providing guidelines for maintaining adequate PAL, which are crucial for developing healthy movement habits in their children [23]. Due to the stay-home recommendations during the pandemic, youth have been spending more time at home with their parents and are under greater parental influence. Therefore, it is important to elucidate/detect the exact parental and familial factors that influence the PALs of youth in order to create the most appropriate interventions to increase PALs during future similar crises.

Collectively, there is evidence that the COVID-19 pandemic has negatively influenced the PAL among adolescents due to imposed measures of social-distancing and lockdown [15,17,18,22]. However, there is a lack of international data regarding the changes in PAL in adolescents, especially taking into account factors influencing PAL and COVID-19-induced changes in PAL. Finally, previous studies noted familial factors as important determinants of PAL, but the problem is understudied with regard to the COVID-19 pandemic [23,25]. As a result, this study aimed to evaluate the dynamics of changes in PAL among adolescents from Bosnia and Herzegovina (B&H) and to evaluate sociodemographic, and parental/familial factors which may influence PAL before and during the COVID-19 pandemic and imposed lockdown.

## 2. Materials and Methods

### 2.1. Participants and Study Design

Participants were 688 adolescents (322 females) from B&H. They were 17 years old at the baseline period of the study (15–18 years of age) and were attending high school. At baseline, all participants were healthy and attended regular physical education classes 2 times per week, and some adolescents also took part in extracurricular sports activities. The sample comprised adolescents residing in three counties, and of the total sample, 65% (445 participants; 202 females) resided in urban centers, and 35% resided in rural communities. Characteristics of the sample are in more details presented in Appendix A. This study is part of another large study (“Physical activity, substance misuse, and factors of influence in adolescence”) which was previously initiated and approved by Ethical Board of Faculty of Kinesiology, University of Split (EBO: 2181-205-05-02-05-14-005); hence, participants were already informed about the study aims, benefits, and risks, and parental consent was collected before the baseline period of this study.

This study involved two testing occasions: (i) baseline testing conducted before the implementation of measures of social distancing due to the COVID-19 pandemic (January 2020) and (ii) follow-up testing conducted during the time when social distancing measures were implemented (late April 2020) (Figure 1).

The baseline testing included sociodemographic-factors, parental/familial factors, and baseline PALs. The follow-up testing only included follow-up PAL measures. It is important to mention that during the baseline testing, adolescents had generally regular routines, they attended school, sports clubs were open, and there were no traveling bans in B&H. However, during the follow-up testing period, measures of social distancing had been imposed, including the closing of schools, sports clubs, fitness centers, and shopping malls, and public gatherings were restricted. Although the testing was anonymous, to pair the responses in two testing waves, participants were instructed to use anonymous codes for identification purposes. Baseline testing was performed by paper-pen questionnaires as explained in detail previously [26]. Follow-up testing was done using online Google Forms, and participants were contacted by e-mail and asked to participate in the survey while using the identification code used previously for baseline testing. At baseline, the 744 participants were tested, and at follow-up 695 participants responded to questionnaire. However, because of the inconsistency in identification codes, 7 participants of those tested at follow-up were not included in this study, altogether resulting in total sample of 688 participants and retention rate of 92%.

### 2.2. Variables

Variables included in this study were basic sociodemographic variables (age and gender), familial/parental factors (predictors), and PALs (criteria).

Familial/parental factors consisted of questions about paternal and maternal education level (university degree, college degree, high school, elementary school) and the financial status of the family (under average–average–above average), as well as responses to the following questions: (i) “How often do you have a conflict with your parents/family?” (never–rarely–from time to time–regularly/frequently); (ii) “How often are your parents/family members absent from home, including for their work obligations?” (never–rarely–from time to time–regularly/frequently); (iii) “How often do your parents/family members ask you questions about your friends, scholastic achievements, problems, and other personal issues?” (never–rarely–from time to time–regularly/frequently); and (iv) “How would you rate how much your parents/family care about you and your personal life?” (Very poor care–Low care–My parents/family care about me–My parents/family care about me a lot). The variables were previously applied and found to be reliable and valid in evaluation of the familial/parental factors in similar samples [26].

The Physical Activity Questionnaire for Adolescents (PAQ-A) was used to assess PALs at baseline (baseline-PAL), and at a follow-up measurement period (follow-up-PAL). The PAQ-A has been demonstrated as reliable and valid in a sample of adolescents from Croatia and Bosnia and Herzegovina [8,22]. PAQ-A is a 7-day recall and self-administered questionnaire that was developed to measure the PAL of adolescents aged 14 to 19 years. Item 1 assesses physical activity during spare time (e.g., bicycling, walking, running, dancing, football); Item 2 assesses physical activity during physical education classes; Item 3 assesses physical activity during lunch break; Item 4 assesses physical activity right after school; Item 5 assesses physical activity during the evenings; Item 6 assesses physical activity during the weekend; Item 7 assesses general physical activity during free time (“describes you best”); Item 8 questions the involvement in physical activity on each day of the week; and Item 9 is used to identify participants who are sick, injured, or have any other cause for reduced physical activity and is not used in the final score. Items 1 to 8 are scored on a scale from 1 to 5, with 1 representing no activity or a low activity level and 5 representing a high activity level. The final theoretical PAQ-A score is calculated as the arithmetic mean of the scores from the first 8 items [27]. Apart from the raw PAQ-A results, for the purpose of statistical analyses in this study (details described later in the paper), the results of the baseline-PAL and follow-up-PAL were dichotomized, and scores below 2.73 were considered low-level-PAL, while scores above 2.73 were considered appropriate-PAL, as suggested in previous studies [28,29].

### 2.3. Statistical Analyses

The normality of the distribution was checked with the Kolmogorov–Smirnov test. Afterwards, means and standard deviations were calculated for PAQ-A (PAL) and age, while percentages and frequencies were calculated for other variables.

Differences between baseline- and follow-up-PAL for the sample as a whole and separately for boys and girls were identified by *t*-tests for dependent samples. Differences between boys and girls for PAL results and age were calculated by *t*-tests for independent samples, and for remaining variables by Mann Whitney U test.

To identify associations between predictors (sociodemographic- and parental/familial-variables) and dichotomized PAL-criteria, logistic regressions were calculated, with Odds Ratios (ORs) and corresponding 95% CI values reported. Since girls were slightly older than boys and preliminary statistics identified significant influence of age and gender on PAL (please see previous text on participants, and later Results for details) logistic regressions were calculated as crude models (Model 0), and additionally controlled for gender and age as covariates (Model 1). The model fit was checked by the Hosmer Lemeshow test (a statistically significant test indicates that the model does not adequately fit the data). A p-value of 0.05 was applied, and the statistical package Statistica ver. 13.5 (Tibco Inc., Palo Alto, CA, USA) was used for all calculations.

## 3. Results

The PAL decreased significantly from baseline to the follow-up testing period in the sample as a whole (from 2.98 ± 0.71 to 2.31 ± 0.68; *t*-test: 11.88, *p* < 0.001), as well as when observed separately among girls (2.69 ± 0.49 to 1.95 ± 0.56; *t*-test: 8.88, *p* < 0.001) and boys (from 3.12 ± 0.56 to 2.50 ± 0.44; *t*-test: 10.01, *p* < 0.01). Boys had higher level of PAL than girls at baseline (*t*-test: 12.55, *p* < 0.001) and at the follow-up measurement period (*t*-test: 11.99, *p* < 0.001) (Figure 2).

Girls were slightly older than boys (17.92 ± 1.00 and 16.96 ± 0.98, *t*-test: 1.66, *p* = 0.048). Differences between boys and girls in studied predictors (sociodemographic-, parental/familial-variable) are presented in Appendix A. Boys and girls differed significantly in most of the studied variables, including: paternal education (higher level reported in boys), maternal education (higher in boys), self-estimated conflict with parents/family (higher in girls), parental/familiar questioning (higher in girls), and parental/familiar care (higher in girls).

Correlates of baseline-PAL are presented in Figure 3. Since crude logistic regression provided evidence of significant influences of gender and age on baseline-PAL, with a higher likelihood of sufficient PAL among boys (OR: 2.50, 95%CI: 1.54–3.03) and lower likelihood of sufficient PAL among younger participants (OR: 0.81, 95%CI: 0.66–0.99), we briefly present only the results of logistic regression model controlled for gender and participants’ age as covariates (Model 1). Paternal education was associated with sufficient baseline-PAL (OR: 1.40, 95%CI: 1.10–1.77), while a lower likelihood of a sufficient level of PAL at baseline was found for adolescents who reported a higher level of conflict with parents/family members (OR: 0.72, 95%CI: 0.57–0.90).

Male gender was positively related to follow-up-PAL, with higher likelihood for sufficient PAL in boys (OR: 2.41, 95%CI: 1.11–4.01). When gender and age were included as covariates in the regression analysis (Model 1), significant correlations were found between familial conflict and follow-up-PAL (OR: 0.77, 95%CI: 0.60–0.99), with a lower likelihood of sufficient PAL at the time of follow-up among adolescents who reported a higher level of conflict with their parents/family. Additionally, adolescents whose fathers were better educated were more likely to achieve a sufficient PAL during the COVID-19 pandemic and the imposed rules of social-distancing (OR: 1.33, 95%CI: 1.19–2.01) (Figure 4).

## 4. Discussion

The most important findings of this study are as follows: (i) PALs significantly decreased during the COVID-19 pandemic, (ii) paternal education was associated with PALs before and during the pandemic, and (iii) conflict with parents/family was a factor that decreased the PAL before and during the pandemic.

### 4.1. Changes in Physical Activity Levels

The decrease in the PAL as a result of the COVID-19 pandemic and imposed measures of social distancing was expected and already confirmed [15,30,31]. Very recent studies conducted on adolescents from Croatia and Italy have registered significant declines in their PALs [17,18,22]. Specifically, while using the same measurement tool as the one applied here, a Croatian study reported a decrease from 2.99 ± 0.70 to 2.67 ± 0.60 [17]. Therefore, we may highlight that the decline in PALs for adolescents from B&H is somewhat more evident (from 2.98 ± 0.71 to 2.31 ± 0.68). There are two plausible explanations for differences between Croatian adolescents and the B&H adolescents studied here: the first is related to differences in the epidemiological situation, and the second is related to the geographical location and differences in climate between the two countries.

First, according to official reports, the epidemiological situation related to the COVID-19 pandemic that occurred in B&H was less favorable than the epidemiological situation in Croatia. Precisely, in B&H, at the time of writing, there have been more than 12,000 confirmed cases of COVID-19 disease, while in Croatia (country with similar population of approximately 4 million residents), there have been less than half as many cases (about 5300 confirmed cases) [32]. Consequently, measures regarding the pandemic in B&H, including movement restrictions, were more rigorous. The more rigorous lockdown measures resulted in greater precaution among people which led to minimized time spent outside of homes, altogether resulting in an even greater decline in PAL for B&H adolescents than for their Croatian peers.

Second, it has been documented that the PAL varies based on region, climate, weather, and season [33]. Specifically, children and adolescents who live in regions with warmer temperatures spend more time outdoors and have higher PALs [33]. Although Croatia and B&H share a long border, B&H is a country with many mountain massifs and valleys and where a temperate-continental climate prevails [34]. This study sampled adolescents from various territories, including continental and mountain parts of B&H. On the other hand, comparative Croatian studies have been conducted in coastal areas with a mild Mediterranean climate [17,22]. Therefore, adolescents observed in recent Croatian studies have been exposed to a mild Mediterranean climate (temperature mostly higher than 10°), and consequently, the weather conditions during the studied period of COVID-19 pandemic were very favorable, so adolescents were able to spend some time outdoors participating in recreational activities while respecting the mandatory measures of social distancing [22]. Collectively, it is likely that the adolescents from B&H studied herein experienced greater declines in PAL than adolescents from Croatia because of (i) the stricter restrictions and measures of social distancing and (ii) the less favorable climate, which together led to lower PALs during the COVID-19 lockdown.

### 4.2. Paternal Education and Physical Activity Levels

One of the most important findings of our research was that paternal education was associated with the PAL in both testing waves, with a higher PAL in children whose fathers were better educated. Before explaining specific influence of paternal education status on the PAL of their children, a brief overview of parental influence on a child’s PAL is needed. Generally, it is relatively well documented that fathers have a more pronounced influence on the sport-participation and physical activity of children than mothers [35]. Almost 30 years ago, Moore et al. established that children with more active fathers have a 3.5 times greater likelihood of being physically active in comparison to children with inactive fathers [36]. This was confirmed later by Yang et al., who reported that the PAL of fathers is significantly correlated with their children’s PAL [37].

Hence, it is considered that children whose fathers are more active are more likely to have greater participation in sports activities than children who have inactive fathers, and the influence of the father is considered to be an important socializing agent for children in relation to sports activities [37]. Supportively, a review study by Beets et al. noted that fathers more frequently initiate and engage in physical activity with their children and use physical activity as a way of socializing and bonding with children [38]. Nowadays, it is accepted that fathers use their own physical activity patterns to directly influence their children’s physical activity habits [39]. With regard to differences between maternal and paternal influence, two issues should be contextualized. First, it is deemed that fathers provide a better example and model of sports skills, since men are generally more involved in sports activities than women/mothers [40]. Second, it seems that maternal influence on their children’s behavior decreases with age, while the father–child relationship is relatively stable in terms of the PAL [23].

While previous discussion has explained the paternal influence on children’s PAL, the specific influence of paternal education level is probably a result of the following factors: first, better paternal education might imply higher consciousness and knowledge of the health benefits of frequent and adequate physical activity. Supportively, previous research has recorded that a higher paternal level of education is associated with better health status of children [41]. Therefore, it is likely that better educated fathers are aware of the benefits of proper PAL during childhood and adolescence and are concerned about the PAL of their children [42]. Second, higher paternal education can lead to a favorable occupational status that contributes to a better financial status, while a better financial status of the family provides appropriate resources for allowing children to participate in organized sports activities [43].

### 4.3. Familial Factors and Physical Activity Levels

Family conflict was found to have a negative association with PALs among adolescents in both testing waves (before and during COVID-19 pandemic). Although these results are not surprising, the background should be specifically explained given the overall importance of PAL in adolescence. Under normal circumstances, parents and children usually spend a lot of time together. Parents determine and influence their children’s life habits [44], and this logically transfers even to physical activity habits [23]. More specifically, parents have a crucial role in the development of the social customs of their children. Therefore, parental activity habits and support are major determinants of children’s physical activity behaviors [45], and such parental influence can be both direct and indirect, influencing the social-cognitive, socio-economic, physical, and cultural environments [46].

There are several main aspects of parental influence on children’s PALs [47]. The first one is role modeling which represents the efforts of parents to be active and their interest in physical activity [21]. Concerning social-cognitive theory, modeling is thought to promote observational learning and provide information about what is important and expected [48,49]. Parental role modeling, therefore, represents a parent’s activity patterns, attitudes, and efforts to model movement behaviors that their children observe and possibly imitate [50]. Role modeling can also include active involvement of parents in physical activity with their children (co-physical activities) which promotes healthy behavior of both the child and parent [51].

In the already-mentioned review by Gustafson et al., it was stated that parental PALs are positively correlated with increases in children’s PAL [50]. It is considered that children whose parents are more active are more likely to be sufficiently active [50]. Additionally, Fuemmeler et al. recorded that the number of active parents influences children’s PALs [52]. More precisely, higher attained PALs were found to be achieved by children whose parents were both active, somewhat lower PALs were observed in children who had only one active parent, and the lowest PALs were recorded for children whose parents were both inactive [36,52].

The second aspect of parental influence on children PALs is parental support, which consists of parental encouragement, involvement, and facilitation [21]. Specifically, parental encouragement was positively correlated with various aspects of physical activity (e.g., the intensity of physical activity, attraction to sports, intention to be active, perceived benefits from physical activity, perceived competence, sports participation, and amount of PA) and is considered to be one of the key determinants of children’s PAL [38,53]. Furthermore, parental involvement, which represents a more overt level of parental support that includes playing with the child and coaching, has a positive influence on children’s PALs [50]. Finally, facilitation represents providing access and opportunities for activity, transportation to sports activity, payment of the club fees, equipment provision, and providing opportunities for outdoor recreational activities [50].

According to Welk et al., parental facilitation is the most significant predictor of children’s involvement and interest in physical activity [47]. A child’s participation in sports activities usually demands on parental financial support; therefore, it is logical that youth largely depend on this kind of parental support [54]. It is expected that more active parents will provide more support for sports involvement, provide more sports equipment, and will be personally involved in more frequent outdoor family time activities, through which they can also influence children’s physical activity habits [38,50]. Collectively, it is unquestionable that parental influence in forms of role modeling, social support, and social influence affect a child’s attraction to physical activity and their perceived competence, through which a child’s PAL can be influenced [47]. In challenging situations such as the COVID-19 pandemic, all of the previously explained factors are probably even more pronounced.

In order to explain why family conflict has been found to negatively influence PALs, family cohesion, the opposite phenomenon, is described. Family cohesion is defined as “emotional bonding that family members have toward one another” [55]. It represents normal family functioning, including caring for other family members, familial support, and affection [56]. Trost et al. observed several constructs that can determine family cohesion: family functioning, family connectedness, family bonding, family control, family expressiveness, and parent–child communication [45]. It is expected that families with better family cohesion will create a more supportive environment for healthy-lifestyle habits, including physical activity habits [45]. Supportively, a positive correlation between family cohesion and children’s PAL has been recorded, which has been explained by the theory that children from better-functioning families get more support and encouragement to be physically active [45]. This was also confirmed by Ornelas et al. who found positive associations of family cohesion, parent–child communication, and parental engagement in physical activity with PAL [57]. Prospective research by Bigman et al. showed a correlation between family cohesion and conflict with PALs among Mexican adolescents [58]. Adolescents who reported high family cohesion had a 32% greater likelihood of having a higher PAL than adolescents who reported lower family cohesion [58]. In contrast, a significant correlation between family conflict and PAL was not found [58].

Collectively, parents who are physically active, and those who support, encourage, and are personally involved in their children’s physical activity, will more likely positively influence their children’s engagement in various types of PAL, irrespective of the situation. This is particularly possible if parents provide children with their own example and create a supportive environment for participation in activity [36]. On the other hand, in situations of conflict between parents and children, the transfer of positive parental behavior (even if such positive behavior exists) to children will not occur.

All of the previously discussed factors probably explain our findings about the negative association between familial conflict and PAL at the baseline and follow-up measurement periods. Specifically, concerning the fact that children and adolescents do not have access to finances and have limited methods of transportation, if parental cooperation does not occur, their involvement in sports activities will be limited. Additionally, the likelihood of spending time in outdoor recreational activities is certainly limited if children are in conflict with their parents, because the parents are the ones who can provide them such activities. It is expected that due to conflict, some children do not have adequate parental influence and, therefore, do not have appropriate support and guidelines for being physically active.

A very recent Canadian National Survey on the Impact of the COVID-19 virus outbreak on movement and play behaviors of Canadian children and youth concluded that as a result of stay-home policies during the pandemic and limited access to sports-recreational facilities, families tried to create new hobbies and activities for maintaining healthy movement habits [25]. Precisely, parents reported that children were more involved in indoor activities, like household chores and activities including dance and physical education exercises, and outdoor activities, like riding a bicycle, walking, playing badminton, street basketball and hockey [25]. The same study established that parental encouragement and co-participation were associated with increases in both indoor and outdoor physical activity [25]. Our results clearly support such considerations, even for adolescents from southeastern Europe.

Naturally, the associations between familial variables and PAL were stronger at baseline than in the follow-up testing period (see Results for details). Indeed, it is likely that all real-life influences appear in regular (i.e., “real-life”) situations. In our case, baseline testing occurred during a period of regular life, when regular physical activities were possible. However, habits developed in such regular situations (pre-pandemic period) are logically translated even to irregular and challenging situations (period of lockdown). Therefore, although correlations obtained between familial factors and PAL at follow up were not so profound, it is still important to point out a certain “transfer of influence”, as already suggested for other indices of overall well-being [17].

### 4.4. Study Limitations and Strengths

The most important limitation comes from the fact that the investigation is based on self-reporting of the data. Therefore, participants may lean toward socially acceptable answers. However, knowing that the study was absolutely anonymous, there is a lower possibility of such bias. In addition, as in any other studies with self-reported data, there is a question of objectivity. While this is possible for some questions (i.e., parental conflict, socioeconomic status), some questions are less likely to be influenced (i.e., parental education). Additionally, this study did not take into account the possibility that some of the studied adolescents had to prepare for the final exam at the end of high school (e.g., maturita exam) and, therefore, probably had less time and opportunities to be physically active. Finally, this study observed adolescents who were involved in regular schooling system and who were able to respond to follow-up questioning using their own technological resources (i.e., smart phones, computers), which almost certainly influenced the participation at both testing waves and presented results. Therefore, associations between scholastic factors and PAL changes during pandemic should be examined in future

This is one of the first studies which investigated the problem of parental influence on PAL during COVID-19 pandemic, and probably the first one which examined the problem in B&H. Results were comparable to previous studies using the same measurement tools. Therefore, although this study is not the final word on this topic, the authors believe that the findings will improve the knowledge about the problem and initiate further studies.

## 5. Conclusions

Family conflict was negatively associated with PALs before and during the pandemic. It is logical to assume that children who generally disagree with their parents will not get adequate support for being physically active, which includes financial support, transport to activity, encouragement, and social support in general. In addition, depression, stress, and anxiety, which commonly occur during home-confinement, probably increased existing familial conflict.

During the pandemic and similar challenging health situations, children are at home with their parents more. Therefore, it is important to develop strategies and methods for sustaining and promoting the PAL among all family members, with special attention on youth. However, during the pandemic, many parents were forced to work from home, and therefore, were more submissive to their children. This increased screen-time and decreased the PAL. Therefore, it is essential to teach parents that they have a key role in influencing children’s PAL and that they should put more effort in the form of support, co-participation, and encouragement to ensure children stay active.

This study clearly highlights that it is of upmost importance to promote the significance and benefits of physical activity. We suggest that educative content developed by health and sports professionals that promotes physical activity should be offered through social networks, online apps, and platforms and should be widely promoted and distributed free of charge or with acceptable fees. Furthermore, solutions for providing similar content for people without access to technology and internet should be developed. Additionally, providing the support to participate in sporting organizations for youth is needed as sport is commonly used for promoting socialization, cooperation, respectful competition, and managing conflict. In addition, the provision of psychological support, ensuring that there are sufficient open spaces and encouraging people to get outdoors and try new outdoor activities as much as possible while respecting social-distancing propositions would be beneficial as well. It is important to note that such activities should be available to everyone in terms of financial, transportational, and security aspects (hand hygiene, maintaining proposed social distancing measures) in order to enable participation.

## Figures and Tables

**Figure 1 children-07-00125-f001:**
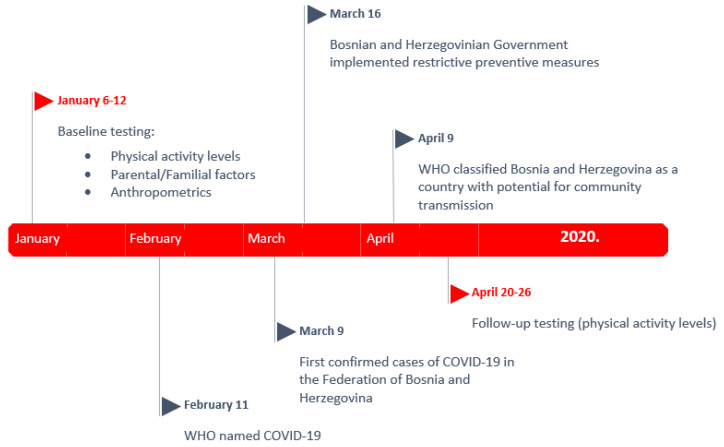
Time line of the investigation with the most important dates considering COVID-19 globally and in Bosnia and Herzegovina.

**Figure 2 children-07-00125-f002:**
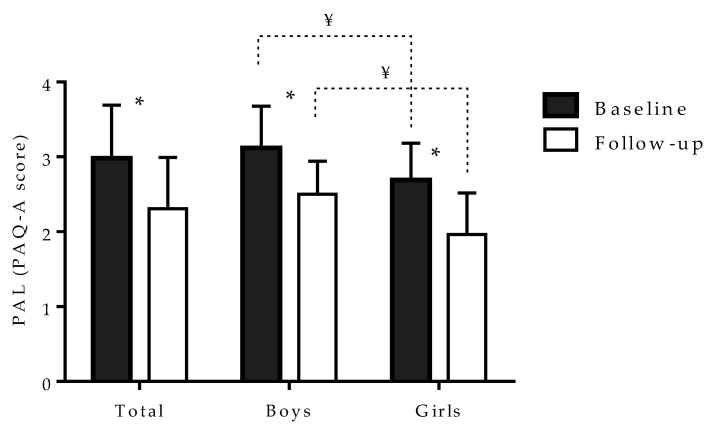
Physical activity levels (PAL) at baseline (before COVID-19 pandemic), and at follow-up (during COVID-19 pandemic) with significant *t*-test differenc1s (¥ indicates significant (*p* < 0.05) differences between groups, * indicates significant (*p* < 0.05) differences within groups).

**Figure 3 children-07-00125-f003:**
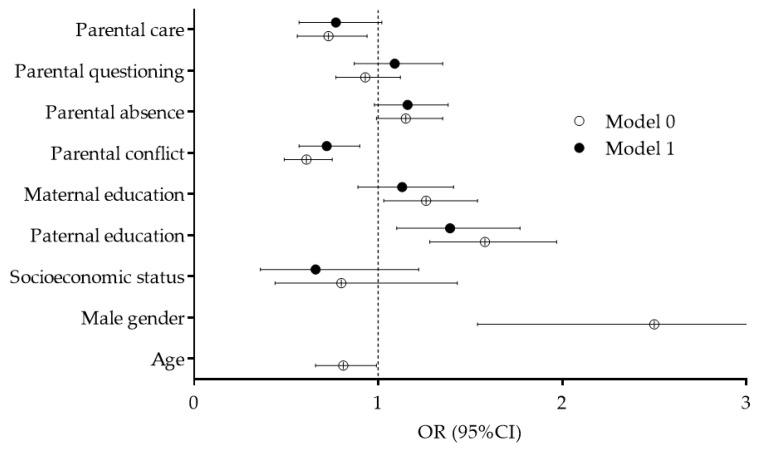
Correlates of sufficient physical activity levels before COVID-19 pandemic (Model 0—crude logistic regression model non-controlled for covariates, Model 1—logistic regression controlled for gender and age).

**Figure 4 children-07-00125-f004:**
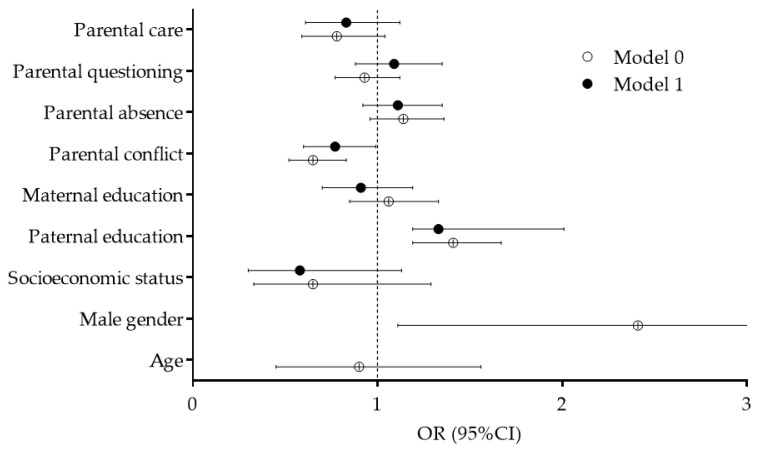
Correlates of sufficient physical activity levels during COVID-19 pandemic (Model 0—crude logistic regression model non-controlled for covariates, Model 1—logistic regression controlled for gender and age).

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
