# Peer review of "Contextualizing Parental/Familial Influence on Physical Activity in Adolescents before and during COVID-19 Pandemic: A Prospective Analysis"

_children, 2020, doi:10.3390/children7090125_

Round 1

Reviewer 1 Report

The manuscript is very well written and the findings are clearly presented.  It would be useful to further develop the findings, discussion, and implications.  The understanding that PA decreased during COVID is not surprising; however, the implications for further gendered impact is as well as the impact of family experiences and conflict.  The authors note the impact for families of this, but it would be useful to expand upon these implications for research and practice.  How can these be used to address future pandemics or such large scale world events?  Lessons learned from this pandemic will be important moving forward. The UN has a policy brief on the impact of COVID and PA that could be used to assist in the implications section. 

https://www.un.org/development/desa/dpad/publication/un-desa-policy-brief-73-the-impact-of-covid-19-on-sport-physical-activity-and-well-being-and-its-effects-on-social-development/

This manuscript has important implications for PA and adolescents especially as we may not have social distancing removed for a long period of time.  

Author Response

The manuscript is very well written and the findings are clearly presented.  It would be useful to further develop the findings, discussion, and implications.  The understanding that PA decreased during COVID is not surprising; however, the implications for further gendered impact is as well as the impact of family experiences and conflict.  The authors note the impact for families of this, but it would be useful to expand upon these implications for research and practice.  How can these be used to address future pandemics or such large scale world events?  Lessons learned from this pandemic will be important moving forward. The UN has a policy brief on the impact of COVID and PA that could be used to assist in the implications section. 

https://www.un.org/development/desa/dpad/publication/un-desa-policy-brief-73-the-impact-of-covid-19-on-sport-physical-activity-and-well-being-and-its-effects-on-social-development/

This manuscript has important implications for PA and adolescents especially as we may not have social distancing removed for a long period of time.

RESPONSE: We are very thankful for your comments and support. This is an emerging topic and we consider that each aspect of influence should be investigated in detail in order to develop strategies for enhancing general well-being and safety during the pandemic and similar crisis. Also, thank you for pointing out the UN’s policy brief, we find it insightful and some considerations have been implemented in the text (Conclusion section, lines 415-421). Text reads: “We suggest that educative content developed by health and sports professionals that promotes physical activity should be offered through social networks, online apps, and platforms and should be widely promoted and distributed free of charge or with acceptable fees. Also, solutions for providing similar content for people without access to technology and internet should be developed. Additionally, providing the support to participate in sporting organizations for youth is needed as sport is commonly used for promoting socialization, cooperation, respectful competition, and managing conflict.”

Staying at your disposal,

Authors

Reviewer 2 Report

Contextualizing parental/familial influence on physical activity in adolescents before and during COVID-19 pandemic: A prospective analysis

This interesting study reacts to the influence of the COVID-19 pandemic (current problem in the world). The pandemic and its resulting precautions have a logical influence on the application of physical activity. Therefore, it is interesting to observe and define variables, which, in connection with the pandemic precautions, influence the participation of adolescents in physical activities. Sufficient application of physical activity is a prerequisite for maintaining health. The authors correctly state that on one side, performing certain physical activities in groups that put adolescents in close proximity to each other increases risk with COVID-19. On the other side, physical activity is associated with an individual’s fitness, healthy development, and the transfer of healthy lifestyle habits into adulthood. The question is, what is the role of parents in this situation. Parents should lead adolescents to a healthy lifestyle and support them in physical activities. However, in adolescence, a parent’s role in supporting their child’s physical activity is declining and other factors are taking over. 

Problems and issues to be appropriately specified and clarified:

Authors state:

They were 17 years old at the baseline period of the study (15–18 years of age) and were attending high school.  Also, they state:  Girls were slightly older than boys (17.92±2.00 and 16.96±1.98, t-test: 1.66, p = 0.048). So, what is the variation range of a sample of girls – it seems that it is too wide and that age of those observed girls is too high. I would say that it should not belong into this monitored sample. What kind of parents’ influence would we like to observe? Even in terms of comparison with boys, I believe the age difference should be discussed.

I’d like to see more precise characteristics of the sample.

It is not clear from the study to what extent parental conflict was defined and how it was detected.

It is clear from the study that data collection in the second phase of the research was handled via a web interface. Was there a change in the number of respondents in relation to the first phase of the research? How were all those families equipped with this option?

Due to the impossibility of submitting the results in the second wave of the research (missing the web interface), were not the social characteristics of the sample defined differently?

It is not clear from the description of those samples how samples selection was defined during the first and second wave of testing.

How could a different way of filling data influence the collected data?

I’d like to see a detailed characterization of living environment of the sample. Did the sample come from a city or a village?  

Is it possible that some of those adolescents were engaged in pandemic-related distance-teaching and that some of them were even finishing their high school and were preparing for their maturita exam? This could indicate that less physical activity may have had more to do with the time of year coinciding with maturita exams. Therefore, other activities than physical ones were preferred. Did authors take all those variables into consideration?

The discussion is qualified, and the conclusion reached quite logical. Although this study has gaps in accuracy of the data obtained, and the definition of the samples, collected and presented data are interesting precisely because of the period – the COVID 19 pandemic. Nevertheless, I would recommend clarifying the data on the questions that have been asked above.

In references, I see inconsistencies in writing magazine titles. I generally recommend checking and making necessary adjustments in the citation list with regard to capital letters. 

Here are the inconsistencies that need to be adjust.

For example:

Fuemmeler, B.F.; Anderson, C.B.; Mâsse, L.C. Parent-child relationship of directly measured physical activity. International Journal of Behavioral Nutrition and Physical Activity 2011, 8, 17; doi:10.1186/1479-5868-8-17.

X

Ornelas, I.J.; Perreira, K.M.; Ayala, G.X. Parental influences on adolescent physical activity: a  longitudinal study. International journal of behavioral nutrition and physical activity 2007, 4, 3; doi:10.1186/1479-5868-4-3.

Similarly:

Lee, S.M.; Nihiser, A.; Strouse, D.; Das, B.; Michael, S.; Huhman, M. Correlates of children and parents being physically active together. Journal of physical activity and health 2010, 7, 776-783; doi:10.1123/jpah.7.6.776.

x

Bigman, G.; Rajesh, V.; Koehly, L.M.; Strong, L.L.; Oluyomi, A.O.; Strom, S.S.; Wilkinson, A.V. Family cohesion and moderate-to-vigorous physical activity among Mexican origin adolescents: a longitudinal perspective. Journal of Physical Activity and Health 2015, 12, 1023-1030; doi:10.1123/jpah.2014-0014

Bedford, J.; Enria, D.; Giesecke, J.; Heymann, D.L.; Ihekweazu, C.; Kobinger, G.; Lane, H.C.; Memish, Z.; Oh, M.-d.; Schuchat, A. COVID-19: towards controlling of a pandemic. The Lancet 2020, 395, 1015-1018; doi:10.1016/S0140-6736(20)30673-5.

X

Hallal, P.C.; Andersen, L.B.; Bull, F.C.; Guthold, R.; Haskell, W.; Ekelund, U.; Group, L.P.A.S.W. Global  physical activity levels: surveillance progress, pitfalls, and prospects. The lancet 2012, 380, 247-257;  doi:10.1016/S0140-6736(12)60646-1.

In conclusion, I believe it would be appropriate to have more information about the specific circumstances of the study participants. In addition, some corrections in the references section is necessary. Subsequently, it would be possible to issue a contribution.

Author Response

Reviewer #2

This interesting study reacts to the influence of the COVID-19 pandemic (current problem in the world). The pandemic and its resulting precautions have a logical influence on the application of physical activity. Therefore, it is interesting to observe and define variables, which, in connection with the pandemic precautions, influence the participation of adolescents in physical activities. Sufficient application of physical activity is a prerequisite for maintaining health. The authors correctly state that on one side, performing certain physical activities in groups that put adolescents in close proximity to each other increases risk with COVID-19. On the other side, physical activity is associated with an individual’s fitness, healthy development, and the transfer of healthy lifestyle habits into adulthood. The question is, what is the role of parents in this situation. Parents should lead adolescents to a healthy lifestyle and support them in physical activities. However, in adolescence, a parent’s role in supporting their child’s physical activity is declining and other factors are taking over. 

RESPONSE: Thank you for recognising the value of our idea to investigate this topic. Physical activity is under great challenge in this fast-modern world and is an issue in itself. Therefore, during the crisis it is logical to expect that it would be even a greater issue. We must agree that parent’s roles and influences are declining during adolescence, but due to pandemic and home-confinement we assumed that adolescents are under greater parental control and influence than during the normal situations. We are very thankful for your comments and suggestions and we tried to amend the manuscript accordingly.

Problems and issues to be appropriately specified and clarified:

Authors state:

They were 17 years old at the baseline period of the study (15–18 years of age) and were attending high school.  Also, they state:  Girls were slightly older than boys (17.92±2.00 and 16.96±1.98, t-test: 1.66, p = 0.048). So, what is the variation range of a sample of girls – it seems that it is too wide and that age of those observed girls is too high. I would say that it should not belong into this monitored sample. What kind of parents’ influence would we like to observe? Even in terms of comparison with boys, I believe the age difference should be discussed. I’d like to see more precise characteristics of the sample.

RESPONSES:

With regard to differences between boys and girls in age:

  • Thank you for this observation and suggestion. Indeed, girls were slightly older than boys, and this is one of the reasons why age was taken into the logistic regression as covariate and is controlled that way (even for 2nd wave where there was no significant partial influence of age on criterion) This is now more detailly explained in the Statistical Analyses section, and text now reads: “To identify associations between predictors (sociodemographic- and parental/familial-variables) and dichotomized PAL-criteria, logistic regressions were calculated, with Odds Ratios (ORs) and corresponding 95%CI values reported. Since girls were slightly older than boys and preliminary statistics identified significant influence of age and gender on PAL (please see previous text on participants, and later Results for details) logistic regressions were calculated as crude models (Model 0), and additionally controlled for gender and age as covariates (Model 1)” (please see text on Statistical analysis in Methods section, thank you)

With regard to more detailed presentation of the study sample

  • In this version of the manuscript we tried to explain the sample of participants more precisely. Also, details on studied sample were presented in supplementary table 1. Text reads: “Participants were 688 adolescents (322 females) from B&H. They were 17 years old at the baseline period of the study (15–18 years of age) and were attending high school. All participants were healthy and attended regular physical education classes 2 times per week, and some adolescents also took part in extracurricular sports activities. The sample comprised adolescents residing in Herzegovina Neretva County, Western Herzegovina County, and Tuzla County, and of total sample 65% (445 participants; 202 females) resided in urban centers, and 35% resided in rural communities. Characteristics of the sample are in more details presented in Supplementary table 1.” (please see 1st paragraph of the Materials subsection, thank you)

It is not clear from the study to what extent parental conflict was defined and how it was detected.

RESPONSE: Parental conflict is assessed trough the following question: “How often do you have a conflict with your parents/family?” It was detected by evaluating provided answers (never – rarely - from time to time - regularly/frequently). The variables are now explained as follows: “Familial/parental factors consisted of questions about paternal and maternal education level (university degree, college degree, high school, elementary school) and the financial status of the family (under average – average – above average), as well as responses to the following questions: (i) “How often do you have a conflict with your parents/family?” (never – rarely - from time to time - regularly/frequently); (ii) “How often are your parents/family members absent from home, including for their work obligations?” (never – rarely - from time to time - regularly/frequently); (iii) “How often do your parents/family members ask you questions about your friends, scholastic achievements, problems, and other personal issues?” (never – rarely - from time to time - regularly/frequently); and (iv) “How would you rate how much your parents/family care about you and your personal life?” (Very poor care – Low care – My parents/family care about me – My parents/family care about me a lot). The variables were previously applied and found to be reliable and valid in evaluation of the familial/parental factors in similar samples [26].  “ (please see 2nd paragraph of the Variables subsection

It is clear from the study that data collection in the second phase of the research was handled via a web interface. Was there a change in the number of respondents in relation to the first phase of the research? How were all those families equipped with this option?

RESPONSE: Thank you for this suggestions and observation. We tried to highlight this issue more precisely, and text reads: “At baseline, the 744 participants were tested, and at follow-up 695 participants responded to questionnaire. However, because of the inconsistency in identification codes, 7 participants of those who responded on those tested at follow-up were not included in this study, altogether resulting in retention rate of 92%. “

Due to the impossibility of submitting the results in the second wave of the research (missing the web interface), were not the social characteristics of the sample defined differently?

and

It is not clear from the description of those samples how samples selection was defined during the first and second wave of testing.

and

How could a different way of filling data influence the collected data?

RESPONSE: Thank you for your comments. Indeed, these problems were possible but unfortunately because of the study design and the fact that COVID-19 pandemic appeared suddenly we were not prepared for all these problems. Observed adolescents were attending regular schooling system and indeed, only those all had their own technology equipment including smart phone and computer/tablet. However, we had no opportunity to evidence whether the possession of the IT equipment was the factor that directly influenced the retention in the study. This issue has been mentioned in the section Study limitations and strengths, and text reads: “Finally, this study observed adolescents who were involved in regular schooling system and who were able to respond to follow-up questioning using their own technological resources (i.e. smart phones, computers), which almost certainly influenced the participation at both testing waves.” (please see last subsection of the Discussion

I’d like to see a detailed characterization of living environment of the sample. Did the sample come from a city or a village?

RESPONSE: Thank you. We included these details in Participants subsection, and text reads: “The sample comprised adolescents residing in Herzegovina Neretva County, Western Herzegovina County, and Tuzla County, and of total sample 65% (445 participants; 202 females) resided in urban centers, and 35% resided in rural communities.”

RESPONSE:

Is it possible that some of those adolescents were engaged in pandemic-related distance-teaching and that some of them were even finishing their high school and were preparing for their maturita exam? This could indicate that less physical activity may have had more to do with the time of year coinciding with maturita exams. Therefore, other activities than physical ones were preferred. Did authors take all those variables into consideration?

RESPONSE: Thank you for this observation. Yes, some of the adolescents were finishing their high school and were probably having greater needs to spend more time studying and preparing for their maturita exam. However, we did not consider this as a factor and it was added it in the Study limitations. Added text reads:Also, this study did not take into account the possibility that some of the studied adolescents had to prepare for the final exam at the end of high school education and, therefore, probably had less time and opportunities to be physically active.”
Also, we will try to investigate it in future studies, thank you for pointing it out.

The discussion is qualified, and the conclusion reached quite logical. Although this study has gaps in accuracy of the data obtained, and the definition of the samples, collected and presented data are interesting precisely because of the period – the COVID 19 pandemic. Nevertheless, I would recommend clarifying the data on the questions that have been asked above.

RESPONSE: Thank you for showing your interest in this study and topic. We tried to clarify and correct the suggested parts of the manuscript.

In references, I see inconsistencies in writing magazine titles. I generally recommend checking and making necessary adjustments in the citation list with regard to capital letters. 

Here are the inconsistencies that need to be adjust.

For example:

Fuemmeler, B.F.; Anderson, C.B.; Mâsse, L.C. Parent-child relationship of directly measured physical activity. International Journal of Behavioral Nutrition and Physical Activity 2011, 8, 17; doi:10.1186/1479-5868-8-17.

X

Ornelas, I.J.; Perreira, K.M.; Ayala, G.X. Parental influences on adolescent physical activity: a  longitudinal study. International journal of behavioral nutrition and physical activity 2007, 4, 3; doi:10.1186/1479-5868-4-3.

Similarly:

Lee, S.M.; Nihiser, A.; Strouse, D.; Das, B.; Michael, S.; Huhman, M. Correlates of children and parents being physically active together. Journal of physical activity and health 20107, 776-783; doi:10.1123/jpah.7.6.776.

x

Bigman, G.; Rajesh, V.; Koehly, L.M.; Strong, L.L.; Oluyomi, A.O.; Strom, S.S.; Wilkinson, A.V. Family cohesion and moderate-to-vigorous physical activity among Mexican origin adolescents: a longitudinal perspective. Journal of Physical Activity and Health 201512, 1023-1030; doi:10.1123/jpah.2014-0014

Bedford, J.; Enria, D.; Giesecke, J.; Heymann, D.L.; Ihekweazu, C.; Kobinger, G.; Lane, H.C.; Memish, Z.; Oh, M.-d.; Schuchat, A. COVID-19: towards controlling of a pandemic. The Lancet 2020, 395, 1015-1018; doi:10.1016/S0140-6736(20)30673-5.

X

Hallal, P.C.; Andersen, L.B.; Bull, F.C.; Guthold, R.; Haskell, W.; Ekelund, U.; Group, L.P.A.S.W. Global  physical activity levels: surveillance progress, pitfalls, and prospects. The lancet 2012380, 247-257;  doi:10.1016/S0140-6736(12)60646-1.

RESPONSE: Thank you for this comment. Reference list has been detailly checked and capital letters of the journal names corrected/added.

In conclusion, I believe it would be appropriate to have more information about the specific circumstances of the study participants. In addition, some corrections in the references section is necessary. Subsequently, it would be possible to issue a contribution.

RESPONSE: Once again, thank you for providing us such valuable suggestions and comments. We hope that we successfully amended the manuscript.

Staying at your disposal,

Authors